# Strong Gene Flow Undermines Local Adaptations in a Host Parasite System

**DOI:** 10.3390/insects11090585

**Published:** 2020-09-01

**Authors:** Perttu Seppä, Mariaelena Bonelli, Simon Dupont, Sanja Maria Hakala, Anne-Geneviève Bagnères, Maria Cristina Lorenzi

**Affiliations:** 1Centre of Excellence in Biological Interactions, Organismal and Evolutionary Biology Research Program, Faculty of Biological and Environmental Sciences, University of Helsinki, P.O. Box 65, 00014 Helsinki, Finland; sanja.hakala@helsinki.fi; 2Department of Life Sciences and Systems Biology, University of Turin, Via Accademia Albertina 13, 10123 Torino, Italy; mariaelena.bonelli@unito.it (M.B.); lorenzi@univ-paris13.fr (M.C.L.); 3Institut de Recherche sur la Biologie de l’Insecte, UMR 7261, CNRS—Université de Tours, Avenue Monge, Parc Grandmont, 37200 Tours, France; simon.dupont@univ-tours.fr (S.D.); ag.bagneres@cefe.cnrs.fr (A.-G.B.); 4Centre d’Ecologie Fonctionnelle et Evolutive, CNRS UMR5175, Université Montpellier, Université Paul Valery Montpellier 3, EPHE, IRD, 34293 Montpellier, France; 5Laboratory of Experimental and Comparative Ethology (LEEC), University of Sorbonne Paris Nord, 93430 Villetaneuse, France

**Keywords:** AFLP, dispersal, DNA microsatellites, gene flow, genetic diversity, local adaptations, mtDNA, *Polistes*, social parasitism, spatial genetic structure

## Abstract

**Simple Summary:**

The co-evolution of hosts and parasites depends on their ability to adapt to each other’s defense and counter-defense mechanisms. The strength of selection on those mechanisms may vary among populations, resulting in a geographical mosaic of co-evolution. The boreo-montane paper wasp *Polistes biglumis* and its parasite *Polistes atrimandibularis* exemplify this type of co-evolutionary system. Here, we used genetic markers to examine the genetic population structures of these wasps in the western Alps. We found that both host and parasite populations displayed similar levels of genetic variation. In the host species, populations located near to each other were genetically similar; in both the host and the parasite species populations farther apart were significantly different. Thus, apparent dispersal barriers (i.e., high mountains) did not seem to restrict gene flow across populations as expected. Furthermore, there were no major differences in gene flow between the two species, perhaps because *P. atrimandibularis* parasitizes both alpine and lowland host species and annually migrates between alpine and lowland populations. The presence of strong gene flow in a system where local populations experience variable levels of selection pressure challenges the classical hypothesis that restricted gene flow is required for local adaptations to evolve.

**Abstract:**

The co-evolutionary pathways followed by hosts and parasites strongly depend on the adaptive potential of antagonists and its underlying genetic architecture. Geographically structured populations of interacting species often experience local differences in the strength of reciprocal selection pressures, which can result in a geographic mosaic of co-evolution. One example of such a system is the boreo-montane social wasp *Polistes biglumis* and its social parasite *Polistes atrimandibularis*, which have evolved local defense and counter-defense mechanisms to match their antagonist. In this work, we study spatial genetic structure of *P. biglumis* and *P. atrimandibularis* populations at local and regional scales in the Alps, by using nuclear markers (DNA microsatellites, AFLP) and mitochondrial sequences. Both the host and the parasite populations harbored similar amounts of genetic variation. Host populations were not genetically structured at the local scale, but geographic regions were significantly differentiated from each other in both the host and the parasite in all markers. The net dispersal inferred from genetic differentiation was similar in the host and the parasite, which may be due to the annual migration pattern of the parasites between alpine and lowland populations. Thus, the apparent dispersal barriers (i.e., high mountains) do not restrict gene flow as expected and there are no important gene flow differences between the species, which contradict the hypothesis that restricted gene flow is required for local adaptations to evolve.

## 1. Introduction

A close association of species which instigate a selection pressure on each other is one of the main drivers of evolution, particularly when the interaction is antagonistic [1,2,3]. Coevolution between such species leads to an escalation of defense and counter-defense mechanisms and species engaged in a coevolutionary arms race are required to continuously adapt to recurrently evolving mechanisms in order to compete with their opponent [4,5] and therefore need to have a high adaptive potential [6]. As parasites generally have larger populations, shorter generation times, and higher mutation and migration rates compared to their hosts, they typically have more genetic diversity in their populations [7] and as a consequence, they are expected to be ahead in the co-evolutionary arms race [6,8].

Spatial population structure is an important factor influencing evolutionary outcomes, including between-species interactions [9]. Gene flow among populations both decreases population differentiation and can effectively mitigate the loss of genetic diversity due to genetic drift in small and isolated populations. One outcome of gene flow is that it affects the fate of locally adapted traits. In classical one or two locus population genetic models, gene flow has been seen as a force that prevents the evolution of local adaptations, or eradicates the existing ones, unless selection is strong enough to maintain advantageous alleles [10,11]. However, local adaptation has been shown to emerge and be maintained even under gene flow. First, gene flow increases additive genetic variation in quantitative traits by introducing beneficial alleles to the population, consequently enhancing local adaptations [12]. Second, when genetic architecture behind the traits evolving during the co-evolutionary arms race is complex, local adaptations are possible under gene flow when recombination around the genes controlling the traits is reduced [13,14]. Finally, geographically structured populations of interacting species often experience local differences in the strength of reciprocal selection pressures, resulting in a geographic mosaic in these traits. The maintenance of a geographic mosaic entails a dynamic reciprocal change of traits across landscapes, including gene flow among populations [3,15].

Primitively eusocial wasps that lack morphological castes are a long-standing model system for studying the evolution and maintenance of sociality in general e.g., [16] and they are also considered optimal systems for studying social parasitism and co-evolutionary dynamics in host-parasite interactions e.g., [17]. Obligate social parasites (inquilines) lack a worker caste and utilize the social life style of their host species instead of rearing their own offspring. As in all parasitic systems, social parasites and their hosts are engaged in a coevolutionary arms race, but social parasite systems have been suggested to lack the asymmetry in evolutionary potential that is often found in other types of host-parasite associations (e.g., different effective population sizes and generation times) [6], making the prediction which antagonist should be ahead in the co-evolutionary arms race less straightforward [18,19]. Parasitism reduces the fitness of the host [20,21], which imposes a strong selection pressure on the host to fend off parasites [5]. Local adaptations for avoiding parasites have been described in a few host-parasite systems [22,23,24].

*Polistes biglumis* is a boreo-montane social wasp [25], with a western palearctic distribution [25,26,27]. In Southern Europe, it occurs in alpine meadows at elevations between ≈1200 and ≈2000 m a.s.l. Its distribution is patchy, and populations are separated by important geographical barriers, such as high mountains, rocky areas, and forests [24]. Some *P. biglumis* populations are regularly parasitized by an obligate inquiline *P. atrimandibularis*, which occurs around the Mediterranean and Caspian basins [26]. *P. atrimandibularis* is a generalist that parasitizes several *Polistes* hosts [28], but the other host species do not occur sympatrically with *P. biglumis* [24].

Co-evolving interactions between *Polistes* social parasites and their hosts have been studied relatively well in the *biglumis-atrimandibularis* system [17,24,29,30,31,32,33,34]. Previous studies have shown that *P. biglumis* populations differ in their life-history traits and chemical profiles and that the divergence between populations may be ascribed to the selection pressure instigated by the social parasite [24,35]. These local adaptations include, for instance, smaller brood investment and larger protection effort of the host in parasitized compared to non-parasitized populations [24]. Likewise, *P. atrimandibularis* has evolved locally to use several behavioral and chemical strategies to take over the host colony [29,35,36,37,38,39], and mediate predation of host nests [24].

Our primary goal in this work was to test the classical hypothesis that isolation of populations is a prerequisite for local adaptation to evolve in species engaged in a co-evolutionary arms race. For this end, we extended and refined the earlier work by Bonelli et al. [40] to examine in more detail spatial population structure in the paper wasp, *P. biglumis*, by adding another mitochondrial gene and analyzing all samples also in nuclear markers. Moreover, we included similar data on the parasite *P. atrimandibularis*. As topography of our study area is dominated by high mountains and local adaptation has been found in both species [24,31,34,41], we predict that both host and parasite populations are spatially structured, with limited gene flow at both local and regional scales. Only part of the host populations are parasitized by *P. atrimandibularis* and only part of the nests in infested host populations are parasitized and, moreover, reproduction is dominated by a single female in both host and parasite nests [24,42,43]. Thus, parasite populations are presumably relatively small and more widely spaced compared to the host populations [44], and we expect parasite populations to be genetically more differentiated from each other than the host populations are.

We also tested which of the antagonists has a higher adaptive potential, by assessing how much genetic variation the host and parasite populations harbor in our limited sample of neutral markers. We hypothesize that genetic variation is smaller in the parasite compared to the host populations, because the effective population size is presumably smaller and genetic drift stronger in the parasite compared to the host populations (see above).

More precise predictions on the distribution of genetic variation in the study system must be based on males’ and young females’ dispersal propensity and capability, however, and they are partly contradictory. On one hand, *Polistes* females are supposedly philopatric [45], which is indicated by the close relatedness among co-foundresses in practically all *Polistes* species studied so far e.g., [46,47,48,49]. Furthermore, there are also indications of males mating locally in *P. biglumis* [42], i.e., that also male gene flow would be restricted. On the other hand, wasp females are powerful fliers and can be expected to manage long-range dispersal e.g., [50]. In particular, new females and males in *Polistes* inquilines have been suggested to migrate from their nesting sites to hilltops (“hill topping”) in the end of the summer, where they mate and remain until spring [51]. If *P. atrimandibularis* extended its migration to the mountain areas where *P. biglumis* occurs, it would connect the parasite population as a larger entity and should lead to weak population structure across large areas. Thus, the outcome of dispersal depends on the fractions of males and females that are philopatric and that which are engaged in dispersal.

## 2. Materials and Methods

### 2.1. Sampling

Our sampling design is based on extensive knowledge of our study area in the Italian Alps and bordering areas in France and Switzerland by one of the authors (M.C.L.), stemming from surveys spanning more than three decades. In 2005 and 2011–2013, we sampled *P. biglumis* from five geographic regions (major valleys), separated by high mountains. Two of the valleys were located in the south-western part (Valli del Cuneese, Val di Susa), one in the central (Valsesia), and two in the north-eastern part (Val Leventina, Valtellina) of the study area (Figure 1). We sampled one wasp from 1–48 nests (total 215 wasps) from 1–7 local populations (total 20 populations) within these five regions. Distances between local populations ranged from 3 to 58 km and from 72 to 305 km between centroids of the geographic regions. We sampled 1–17 *P. atrimandibularis* females (total 33 wasps) from *P. biglumis* nests in five local populations in three of the regions above (see Appendix A for details of sampling and sampling locations). Part of the samples was previously used by Bonelli et al. [40].

### 2.2. Molecular Analyses

Wasps were analyzed using six DNA microsatellite loci previously developed for *P. dominula* [52], see also [49]. We also generated amplified fragment length polymorphism (AFLP) fragments for all sampled wasps using standard protocols modified from Vos et al. [53] and following Whitlock et al. [54] for data normalization, filtering, locus selection, and phenotype calling. Finally, we sequenced portions of the mitochondrial COI and 16S genes for 130 *P. biglumis* and 24 *P. atrimandibularis* wasps. All details of laboratory procedures are given as a Appendix A.

### 2.3. Population Genetic Analyses

In each local population, we tested random mating (deviation from Hardy-Weinberg equilibrium) in each locus and linkage disequilibrium in each locus pair, and described the amount of genetic variation as allele count, allelic richness (*A*_R_, [55]) and unbiased gene diversity (*H*_E_, [56]). For the AFLP and mitochondrial sequence data, we calculated unbiased haploid gene diversity (*h*_AFLP_, *h*_mt_) and for the latter also nucleotide diversity (π). We tested the difference in diversity measures in the host and parasite populations, as well as in parasitized and non-parasitized host populations with Mann-Whitney tests in IBM SPSS v. 25 [57]. Due to the uneven sample sizes in the two study species, we compared the cumulative haplotype count using rarefaction [58].

We assessed spatial population structure in three ways. First, we partitioned genetic diversity and estimated associated fixation indices in nuclear (*F*) and mitochondrial (*Φ*) markers with AMOVA [59]. In *P. biglumis*, geographic regions were the higher and local populations the lower intermediate levels in the hierarchical model, respectively. Next we concentrated on the regions where both the host and the parasite were sampled (see Table 1), performing an AMOVA by combining samples as three regions. We disregarded local populations as a hierarchical level in this analysis, because of the small sample size in the parasite. Finally, we tested whether the parasitized *P. biglumis* females in the Ferrere and Montgenèvre populations were genetically differentiated subsets of their respective populations, by estimating genetic differentiation between parasitized females from the rest of the population. AMOVAs were performed with locus by locus model and significance of *F* or *Φ* to be larger than zero is based on permuting individuals across populations and regions for 999 times.

Second, we assessed isolation-by-distance (IBD) by calculating pairwise *F*_ST_ and *Φ*_ST_ values between all pairs of populations and calculating the matrix correlation between genetic differentiation and geographic distance between the sampling sites [60]. We transformed the pairwise *F*_ST_ estimates to *F*_ST_/(1 − *F*_ST_) for linearity [60] and used a natural logarithm of the geographic distances. The significance of the matrix correlation was assessed using the Mantel test (999 permutations, [61]). Population genetic analysis were made by using the software Arlequin 3.5.2.2. [62], Fstat 2.9.3.2 [63], GenAlEx 6.5 [64], and the online version Genepop on the Web [65].

Third, we made a model-based Bayesian clustering analysis to assess the number of genetic clusters in the data in all classes of markers, by using software Baps 6.0 [66,67]. In a mixture analysis, Baps aims to cluster samples to groups so that the allele or haplotype frequency differences among the groups are maximized. We first ran the analyses multiple times defining each time an increasing maximum number of clusters (K) that ranged up to the number of sampling sites. Then we chose a smaller range of values around the largest probability of K in the preliminary runs, and reran the analysis in that range five times for each maximum K. As an additional prior, we used spatial information with the nuclear data and population membership with the mitochondrial data.

### 2.4. Phylogenetic Analyses

We analyzed *P. biglumis* and *P. atrimandibularis* mtDNA sequences separately to assess phylogenetic relationships among samples. We first aligned sequences using the ClustalW algorithm [68] implemented in the software Geneious^®^ Pro 5.6.6 and corrected them manually. Concatenated sequences were analyzed using three methods, neighbor-joining (NJ), maximum-likelihood (ML), and Bayesian inference (BI). The NJ method was applied using Phylo_win [69] and the ML method using PhyML [70], while BI was carried out using MrBayes 3.1.2 [71], running it for 5,000,000 generations. The NJ analysis was conducted using the Kimura 2-parameter distance option. MrAIC was used to find an appropriate sequence evolution model for the data (*P. biglumis*: GTRI-HKYI for BI and GTRI for ML; *P. atrimandibularis*: GTRG for BI and GTRI for ML) [72]. No a priori assumptions about tree topologies were made, and analyses were performed using uniform priors. As outgroups, we used COI (GenBank accession number: HQ947784) and 16S (GenBank: GU596651) sequences of *P dominula* for both species and own samples of *P. gallicus* for *P. biglumis*. We reconstructed a minimum spanning network separately for the two species to show evolutionary relationships between haplotypes by using TCS v. 1.2.1. [73].

## 3. Results

### 3.1. Genetic Variation in the Host and the Parasite

#### 3.1.1. Nuclear Markers

Six and five DNA microsatellite loci were polymorphic in *P. biglumis* and *P. atrimandibularis,* respectively (Appendix A). After a standard Bonferroni correction for multiple tests, none of the DNA microsatellite locus pairs showed significant linkage disequilibrium (LD) in *P. atrimandibularis*, but one locus pair did in one *P. biglumis* population (Pdom1 and Pdom25 in Fer). As LD was restricted to just one locus pair in one population, we retained all the loci in the analysis in both species. After correcting for multiple tests, one locus in one population (Pdom1 in Montgenèvre) in *P. biglumis* was not in Hardy-Weinberg equilibrium and this also caused the combined Chi-square value across loci for that population to significantly deviate from random mating. Again, we retained all the loci in the analysis (Appendix A).

After data normalization, filtering, locus selection, and phenotype calling we ended up with 77 and 76 polymorphic AFLP markers in *P. biglumis* and *P. atrimandibularis*, respectively. *P. biglumis* data included 127 unique profiles, of which only 26 (20%) were shared by multiple individuals and seven (6%) were found within the same geographic region. In *P. atrimandibularis*, we found 25 different profiles, of which four (16%) were shared by multiple individuals and two of these were found within the same geographic region.

Our host populations had similar levels of genetic variation in both nuclear markers (Table 1). On average, parasite populations had larger gene diversity in the DNA microsatellite loci compared to the host populations (Mann-Whitney test, *p* = 0.032), but the levels of genetic variation were similar in the allele count and richness, as well as in all measures of genetic variation in AFLP markers (Table 1; Mann-Whitney tests, all *p*’s > 0.19).

#### 3.1.2. Mitochondrial Sequences

Of the mitochondrial genes, we used a total of 1030 bp in *P. biglumis* (16S: 577 bp, COI: 453 bp) and 1037 bp in *P. atrimandibularis* (16S: 577 bp, COI: 460 bp). In *P. biglumis*, we detected 21 substitutions and a total of 18 haplotypes, of which 10 were singletons and eight found in the western part of our study area. In the minimum spanning network, the main clade of haplotypes (clade 1, Figure 2) was located around three central and common haplotypes (h2, h8, h15), with the less common haplotypes differing from them by 1–3 substitutions. The rest of the haplotypes formed three clades separated by 4–5 substitutions from the main group and/or from each other. In *P. atrimandibularis*, we detected five substitutions and a total of six haplotypes. In the minimum spanning network, haplotypes were separated from a central haplotype (h1) by one to two substitutions (Figure 3). When more than one wasp was sequenced, only a single population in the host (Carì) and a single region in the parasite (Valtellina) were monomorphic. The haplotype count was smaller in *P. atrimandibularis* compared to *P. biglumis* (Appendix A).

In *P. biglumis*, mitochondrial variation was highest in the western end of our study area (Valli del Cuneese + Val Susa: 13 haplotypes), followed by the central (Valsesia: 6) and eastern parts (Val Leventina + Valtellina: 4) and populations in the eastern end of our study area also had clearly lowest haplotype (Valtellina) and nucleotide (Val Leventina, Valtellina) diversities (not tested). Parasite populations had smaller nucleotide and haplotype diversities than host populations, although the difference was not significant in the latter (π: *p* = 0.031; *h*_mt_: *p* = 0.15) and this also applied when comparing regions shared by *P. biglumis* and *P. atrimandibularis* (Table 2, not tested). Finally, there were no differences between parasitized and parasite-free host populations in mean nuclear or mitochondrial diversities (Mann-Whitney tests, all *p*’s > 0.11).

### 3.2. Spatial Population Structure

Our results show that the host is spatially structured only at the level of geographic regions. *F*_ST_ and *Φ*_ST_ values showed that local *P. biglumis* populations were not genetically differentiated within geographic regions in any of the markers, but genetic differentiation among regions was significant (Table 2). Genetic differentiation was weak in nuclear markers in general and in DNA microsatellites in particular. However, the ratio of mitochondrial to DNA microsatellite differentiation (*Φ*_ST-mt_*/F*_ST_) was much larger than three (Table 2), the ratio expected with equal male and female dispersal [74].

Estimates of genetic differentiation (*F*_ST_, *Φ*_ST_) among *P. atrimandibularis* regions were larger than among corresponding *P. biglumis* regions in DNA microsatellites and mitochondrial markers, but this may be due to the unbalanced sample sizes in the two species (Table 2). In the AFLP markers, differentiation among regions was also higher in *P. atrimandibularis* than in *P. biglumis*, but the *Φ*_ST-AFLP_ value was not significantly larger than zero in the former (Table 2). Females in parasitized and non-parasitized host nests were not significantly differentiated from each other in any of the markers studied (DNA microsatellites: Ferrere: *F*_ST_ = 0.019, *p* = 0.072; Montgenèvre: *F*_ST_ = −0.015, *p* = 0.99; AFLP: Ferrere: *Φ*_ST_ = −0.040, *p* = 0.66; Montgenèvre: *Φ*_ST_ = −0.011, *p* = 0.31; mtDNA: Ferrere: *Φ*_ST_ = 0.137, *p* = 0.22; Montgenèvre: *Φ*_ST_ = −0.127, *p* = 0.81).

In *P. biglumis*, pairwise *F*_ST_ and *Φ*_ST_ estimates varied widely, often apparently caused by small sample sizes (Appendix A). In DNA microsatellites, the northernmost population Carì appeared to be most distinct from the rest of the populations, with 13 of 17 possible pairwise *F*_ST_ values being significantly larger than zero. There were no significant pairwise *Φ*_ST_ comparisons in AFLP markers, but in mtDNA markers, populations in Valtellina, Val Leventina (Carì), and Valsesia populations Rima and Sant’Antonio had the largest proportion of large pairwise *Φ*_ST_ comparisons, with 10–15 of 17 possible pairwise comparisons significantly larger than zero.

Matrix correlation between pairwise genetic differentiation and geographic distance (Appendix A) was positive and significant in DNA microsatellites and mitochondrial DNA, but not in AFLP markers (Figure 4), suggesting isolation-by-distance in both nuclear and mitochondrial genomes at the scale of the study area. In mitochondrial data, the highest pairwise *Φ*_ST_ values (e.g., *Φ*_ST_ = 1) inflated the transformed *F*_ST_/(1 − *F*_ST_) values (see Figure 4). As these were most common when one of the Valtellina populations (Trivigno, Campovecchio) was the other member of the pair, we reanalyzed the data also by excluding Valtellina populations from the analysis, but IBD remained highly significant (*p* < 0.001). In *P. atrimandibularis*, only three regions were sampled and no formal IBD analysis was made, but in DNA microsatellites and mitochondrial sequences the largest pairwise *F*_ST_ and *Φ*_ST_ values were found between regions where the region furthest apart (Valtellina), was the other member of the pair (Appendix A).

In *P. biglumis*, Bayesian clustering analysis reflects and refined the pattern emerging from AMOVA. In DNA microsatellites, all *P. biglumis* samples were included in a single cluster (K = 1, *p* = 1) and in the AFLP data, individuals were clustered randomly in nine clusters (K = 9, *p* = 0.99). In the latter, all clusters had members from several regions and samples from all regions were spread to several clusters. In the mitochondrial sequence data, the optimal number of clusters was six (K = 6, *p* = 0.96, Appendix A). Optimal partitioning divided sequences geographically to some extent, as almost all sequences from the eastern part of our sampling area (Val Leventina, Valtellina) clustered together, with about half the samples from the central region (Valsesia). The rest of the sequences from regions Valli del Cuneese, Val Susa and Valsesia were divided randomly into five regions. This suggests ample mixing of haplotypes within western and eastern parts of the main sampling area, and a transitional zone between these two around the Valsesia region.

In *P. atrimandibularis*, individuals clustered to two and three clusters in DNA microsatellite and AFLP data, respectively, with rather high probabilities (DNA microsatellites: K = 2, *p* = 0.85; AFLP: K = 3, *p* = 0.98), but individuals from different geographic regions were mixed in the clusters. In the optimal partition of the mitochondrial sequences, all sequences from Valtellina clustered together with one sample from Valli del Cuneese, and the rest of sequences together (K = 2, *p* = 1).

### 3.3. Phylogenetic Analyses

All tree-building methods resulted in similar trees, and here we show only neighbor-joining trees for the two species (Figure 2 and Figure 3). The trees and the minimum spanning network of haplotypes were consistent with each other and also largely reflected the results on the spatial structure across the study area in nuclear markers. In *P. biglumis,* all well-supported clades (bootstrap values > 50%) included sequences from multiple regions. The most basal bifurcation separated a well-supported clade of sequences from the central and western parts of our study area, which were most distant from the rest of the wasps also in the haplotype network (Figure 2). Another spatially consistent pattern in the phylogenetic tree was that all sequences from Valtellina and Val Leventina belonged to the same clade, but this clade also included sequences from all other regions (Figure 2). In *P. atrimandibularis*, phylogenetic analysis followed Bayesian clustering, by separating the clusters identified with strong support (Figure 3). The rest of the individuals sharing haplotypes were also separated in the tree, but not all with high confidence.

## 4. Discussion

### 4.1. Spatial Genetic Structure

Our results suggest that the host *P. biglumis* forms panmictic populations across areas covering several tens of kilometers and dominated by high mountains. Populations are significantly differentiated from each other only among more distant major geographical areas in an isolation-by-distance fashion. This pattern emerges both from nuclear and mitochondrial markers, and it is corroborated by multiple analyses. In AMOVA, a clearly larger proportion of genetic variation was allocated to the among-regions level than to the among-populations level in all markers, and the corresponding fixation indices were significantly larger than zero only at the among-regions level (Table 2). Bayesian clustering analysis failed to find any clusters in the DNA microsatellite data (K = 1) and made an arbitrary clustering for the AFLP data (K = 9), which both are consistent with ample gene flow among populations. A similar mixed pattern was found in the mitochondrial sequence data in the western part of our study area, but samples from the eastern part tended to cluster together. Our phylogenetic tree based on mitochondrial sequences further confirmed this, with sequences from the eastern part of our study area forming well-supported clades. In *P. biglumis*, differentiation among regions was more pronounced in mitochondrial compared to nuclear (DNA microsatellites) markers (Table 2), suggesting male-biased dispersal at this level. Therefore, an important fraction of the *P. biglumis* males and females must disperse relatively well and particularly males must be able to cover long distances. Similarly, a major geographic barrier (Jamuna-Padma-Upper Meghna river system in Bangladesh) appears to be a dispersal barrier particularly for females in *Polistes olivaceus* [75].

Spatial population structure has attracted relatively little attention in paper wasps, but our results in the *biglumis-atrimandibularis* system are in line with the few existing studies. Significant genetic differentiation has only been found among relatively distant populations (tens to hundreds km apart) [75,76] or across continents [50], but not among nest clusters at local scale e.g., [77,78,79,80,81]. The only earlier genetic study on *P. atrimandibularis* found marginally significant differentiation among individuals (“races”) parasitizing different host species other than *P. biglumis*, but the study was conducted within a small area in Tuscany and did not study spatial population structure at larger geographic scale [82]. The lack of syntopic “races” is interesting, however, because locally adapted traits under strong gene flow has been an enigma [10,11], which begs for further studies on local adaptation in lowland populations of *P. atrimandibularis* and their hosts.

There are also only a couple of examples of spatial population structure in social insect host-parasite systems. Weak differentiation was found both in the temporarily parasitic yellowjacket wasp *Vespula squamosa* and its host *V. maculifrons*, across an area comparable to regions (major valleys) in our study [83]. In contrast, highly significant population differentiation was found in nuclear and mitochondrial markers in slave-making ants and their hosts in two geographic areas, but with inconsistent results concerning whether the host or the parasite populations were more strongly differentiated [19]. Their study areas were also much larger than ours, however, precluding conclusions about shorter-range dispersal. Thus, there are not enough examples to generally conclude which antagonist in social insect host-parasite systems disperses more and, assuming that gene flow allows to better adapt to a changing world [84], is ahead in the co-evolutionary arms race.

#### 4.1.1. Gene Flow and Dispersal

Among regions shared by the host and the parasite, parasite populations were more differentiated both in nuclear and mitochondrial markers compared to the host, but the ratio of mitochondrial to nuclear differentiation was clearly smaller in the parasite. This suggests weaker and less sex-biased gene flow in the parasite compared to the host at this geographic range. However, the smaller effective population size in the parasite should be taken into account when drawing conclusions about gene flow in the two systems. According to Wright’s Island Model of migration [11], genetic differentiation (*F*_ST_) and the number of migrating individuals (*N*_e_*m*) are connected as:$$F_{ST}=\frac{1}{1+4N_{e}m}$$

Solving the equation for the migration rate (*m*) and substituting the estimated *F*_ST_ values for the host and the parasite separately, the effective population size (*N_e_*) of the parasite should be ≈6% of the host population size to reach equal migration rate. This is at the lower limit of the proportion of parasitized nests in infested *P. biglumis* populations (6–24%) [24], suggesting that there are no important differences in dispersal in the two systems. Thus, differences in the *F*_ST_ estimates in this system seem to reflect the strength of the genetic drift rather than differences in dispersal ability and/or propensity.

#### 4.1.2. Selection of the Host Nest by the Parasite

Same *P. biglumis* populations are parasitized year after year by *P. atrimandibularis*, while closely located populations never are [24,31,35]. Moreover, only a fraction of host nests in infested populations are parasitized each year [24,43]. This implies that host nests and populations have certain qualities that make them more susceptible to parasite infiltration, but it has remained unclear what makes certain host nests susceptible.

Cuticular hydrocarbons (CHC) are the most likely mechanism of kin and nestmate recognition in social insects [85], which also applies to recognizing an intruding parasite by the host. As CHC profiles have a heritable component [86,87,88], we also tested whether parasitized *P. biglumis* females are a genetically differentiated subset within their local populations. Based on the non-significant genetic differentiation (*F*_ST_, *Φ*_ST_) between parasitized and non-parasitized females, parasitized *P. biglumis* females were a random sample drawn from their local populations. Thus, parasites seem to target host nests randomly in terms of neutral genetic variation and the infiltration mechanism should be sought elsewhere.

### 4.2. Genetic Variation in the Host and the Parasite

Evolutionary potential of a species or a population depends on the amount of genetic variation it harbors. When two species, such as a host and a parasite, are engaged in a co-evolutionary arms race, the partner that harbors more genetic variation is supposed to be ahead in this race [6,8]. In microparasite systems, this is typically the parasite, with much shorter generation times and much larger effective population sizes compared to their hosts. Permanent social parasites (inquilines) and their hosts have been suggested to lack such asymmetry, however, because inquilines and their hosts are usually closely related (Emery’s rule) and their generation times and population sizes are relatively similar [18]. We predicted that the parasite populations have much smaller effective population sizes compared to the host populations (see above) and, as a consequence, they were expected to harbor less genetic variation. However, the amount of genetic variation in both nuclear and mitochondrial markers (gene diversity, haplotype diversity, nucleotide diversity) was similar in host and parasite populations.

Thus, our results do not confirm our prediction on the difference in genetic diversity in our study species based on their perceived population sizes. Previously similar levels of genetic diversity were found in two geographic areas in two slave making ants and their hosts [19] and in a parasitic wasp (*V. squamosa*) and its host (*V. maculifrons*) [83]. As our work and the examples above cover all types of social parasites (temporary, dulosis, inquilinism) and two main groups of social Hymenoptera (wasps, ants), the lack of asymmetry in evolutionary potential may be universal in social parasites and supports the earlier conclusion that social insect host-parasite systems lack this asymmetry [18,19]. All studies above are based on a limited number of both nuclear and mitochondrial markers, however, and genome-wide studies are needed to confirm these results.

### 4.3. Signatures of Historical Gene Flow

Recently, Bonelli et al. [40] found evidence for separate mitochondrial lineages in *P. biglumis* across Alps, with relatively good concordance with geography and chemical variations of cuticular compounds, but lacking samples from the central parts of our study area (Valsesia) and concentrating on a single mitochondrial gene (COI). One aim of this study was to extend and validate this earlier work [40] by adding another mitochondrial gene (16S) and samples from the central area, and by analyzing all samples also by using nuclear markers. Even though our sampling covered a fraction of its distribution [25,26,27], our mitochondrial results on *P. biglumis* allow us to discuss historical gene flow patterns that may have led to the current haplotype diversity and distribution across our study area.

First, the star-like shape of the haplotype network of the main clade and the current distribution of the haplotypes across the whole study area suggest that these wasps are derived from a common ancestral population in a relatively recent evolutionary past. The plains south of the Alps were probably a suitable habitat for the cold tolerant *P. biglumis* during and after the last glaciation (until about 10,000 years ago), but a subsequent climate change would then have driven *P. biglumis* distribution towards the mountain ranges. The star-like phylogeny of the haplotype network suggests a relatively recent expansion of the population size, which, in this scenario, would have taken place before the distribution of the species became discontinuous. Thus, it is likely that much of the original mitochondrial polymorphisms in *P. biglumis* existed already prior to population separation and the current variation represents a retention of polymorphisms in the ancestral parental stock [89].

Second, due to lack of samples from the central parts of our study area and the lower resolution compared to this work, Bonelli et al. [40] missed the deepest split that singled out several small well-separated clades (h4, h6 + h7, h18). The large genetic distance of the deviating haplotypes suggests an older evolutionary separation of these lineages compared to the ones within clade I. Finding these well-separated haplotypes in our study area may be due to them either being a result of an earlier colonization into the ancestral population from a source that was not sampled for this work, or their direct colonization of the west-central part of our study area. The latter scenario would also require recent dispersal across our study area to explain the current distribution of the haplotypes in the area, but this is consistent with our results on the spatial structure in nuclear markers not precluding rare dispersal events.

*P. atrimandibularis* haplotype network and phylogeny showed a much shallower structure than in the host. The only consistent geographic pattern was the clustering together of the haplotypes sampled from the eastern part of our study area (Valtellina). This suggests a similar historical pattern and colonization of the current distribution of *P. atrimandibularis* as in its host, including the colonization from an unknown source and a more recent dispersal of wasps across our study area, but the unknown source would be more eastern in *P. atrimandibularis*. Based on the current level of population differentiation, we estimated (see above) the relative levels of dispersal in the host and parasite systems to be similar. The precision of this assessment, however, depends on the relative contributions of current and historical levels of gene flow in the two species, which we cannot assess in this work due to the lack of samples outside our study area.

Third, mitochondrial markers in *P. biglumis* were clearly less polymorphic in the eastern compared to the western and central parts of our study area. The haplotypes in Valtellina and Val Leventina were also well-clustered in the haplotype network and phylogenetic tree and the haplotypes were distributed spatially so that the eastern haplotypes (e.g., h14, h15) were largely confined to east-central regions, while other common haplotypes (e.g., h3, h4) were distributed more widely. This suggests either a relatively recent genetic bottleneck in Valtellina and Val Leventina populations or a still another colonization event from an unknown source. Additional haplotypes found in populations not included in this study (Sierra Nevada, Southern Spain; Monte Mare, Central Italy) also indicate that there are other potential sources [own unpublished results].

### 4.4. Assumptions Revisited

In this study, we described the genetic population structure in two paper wasps engaged in a co-evolutionary arms race, with the aim to test the classical population genetic hypothesis about the prerequisite for the evolution of local adaptations. Gene flow turned out to be strong among local host populations and at similar level in the host and parasite systems. We also tested which of the antagonists had more genetic variation, but could not find major differences in the level of nuclear variation between host and parasite populations in the relatively small sample of markers studied. Mitochondrial sequence variation was, however, lower in the parasite compared to the host and showed traces of historical gene flow. Thus, our results largely contradict our predictions derived from the assumptions based on the biology of this host-parasite system, spacing of populations, including the landscape they live in with important dispersal barriers (high mountains), and the assumption of restricted gene flow for local adaptations to evolve. As our findings did not support this prediction, we revisit our assumptions. 

First, arguments for female philopatry in *Polistes* may not apply to *P. biglumis* and *P. atrimandibularis* to a similar extent as to other *Polistes*. *P. biglumis* nests are founded and controlled by a single foundress and reproducing females produced during the previous season do not seek inclusive fitness benefits by teaming up with their natal nestmates to found a nest [24,42,43]. Similarly, the dispersal of females of the inquiline parasite *P. atrimandibularis* is not constrained by the need for co-founding a nest with their relatives [24,43]. Thus, the fraction of philopatric females in neither *P. biglumis* nor *P. atrimandibularis* is constrained by the necessity of co-operative breeding and may be smaller than in most paper wasps.

Second, an important reason to predict significant genetic differentiation among host populations was the topography of our study area. Within the major valleys, the difference in altitude among our local populations was considerable (up to >500 m in some cases) and they were sometimes also separated by mountains much higher (>3000 m) than wasps can conceivably cross, forcing them to seek longer dispersal routes along minor valleys. In *P. atrimandibularis*, wasps are faced with the same geographic constraints. However, the “hill topping” effect [51] may facilitate gene flow by connecting the alpine populations and populations parasitizing lowland host species as a larger entity, but it is not known to which extent the alpine and lowland parasite populations are connected. Parasites are also notoriously difficult to localize in the field and cryptic parasite populations may be more common than anticipated, further facilitating the connections among populations.

Third, current host population sizes may not reflect well their long-term effective sizes. Local extinctions and recolonizations are common in marginal populations [90], causing recurring genetic bottlenecks and/or founder effects. For instance, *P. biglumis* populations may have smaller long-term effective sizes and consequently less genetic diversity compared to what is expected in a stable population system. Conversely stronger gene flow across the parasite populations than expected may increase the effective population size in *P. atrimandibularis* from expected based on current population sizes. Thus, an unexpectedly good long-distance dispersal ability of the host and replenishment of the parasite populations from the lowland and/or cryptic populations could largely explain both the similar level of dispersal across major valleys and the current level of genetic diversity in the host and the parasite. Detailed studies on wasp dispersal would help to understand the dynamics of this host-parasite system better.

Finally, classical population genetic models predict that local adaptations can evolve only under restricted gene flow e.g., [10,11], but such models are simplifications and recent work has shown that local adaptations are possible under gene flow e.g., [2,13,14,15,91,92,93]. Traits related to local adaptations in social wasps are most likely controlled by multiple genes, and both introduction of new genetic material improving local adaptations [12] and linkage disequilibrium and reduced recombination among such genes [13,14] are potential mechanisms allowing local adaptation under gene flow. Moreover, geographic mosaic of coevolution, such as in the *biglumis-atrimandibularis* system [24], is a dynamic system, where local populations experience a continuously changing selection pressure. A geographic mosaic of coevolution entails mixing of traits across space, which, in effect, requires gene flow among populations [4,15]. Thus, local adaptations found in our study system [24,31,34,41] are consistent with the lack of spatial structuring at the regional level, at least in the host, both because the traits in question are most likely polygenic and because they form a mosaic of coevolution [24].

## 5. Conclusions

To conclude, we found that *Polistes* wasps are stronger flyers than anticipated and even parasitic wasps can apparently cover considerable distances, should they opt to disperse from their natal area. Stronger gene flow than expected would explain both the lack of spatial population structure in the host across regional scale and the similar level of genetic diversity in the two systems. Consequently, our results on the *biglumis-atrimandibularis* system actually support the suggested lack of asymmetry in evolutionary potential in social insects [18,19]. Genome-wide studies on genetic diversity, genetic basis of locally adapted traits, and spatial population structure in social insect hosts and parasites on even larger spatial scales are needed to evaluate which party drives the co-evolutionary dynamics in these systems.

## Figures and Tables

**Figure 1 insects-11-00585-f001:**
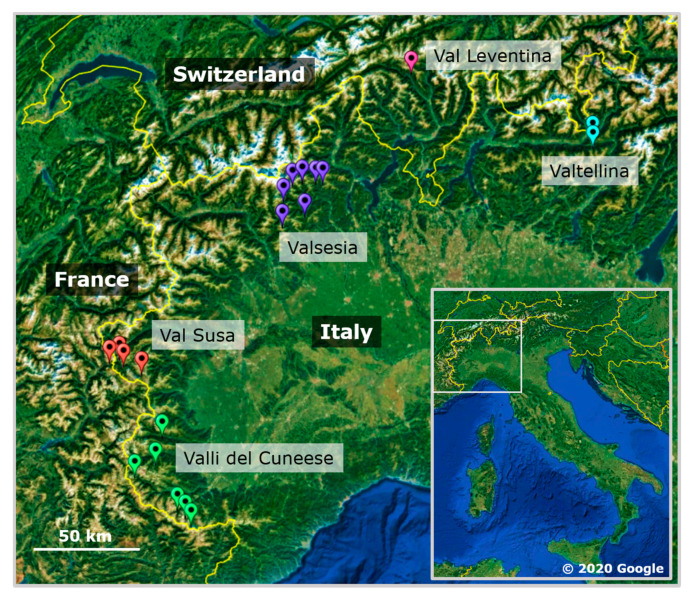
Map of the study area in North-Western Italy and bordering areas in France and Switzerland.

**Figure 2 insects-11-00585-f002:**
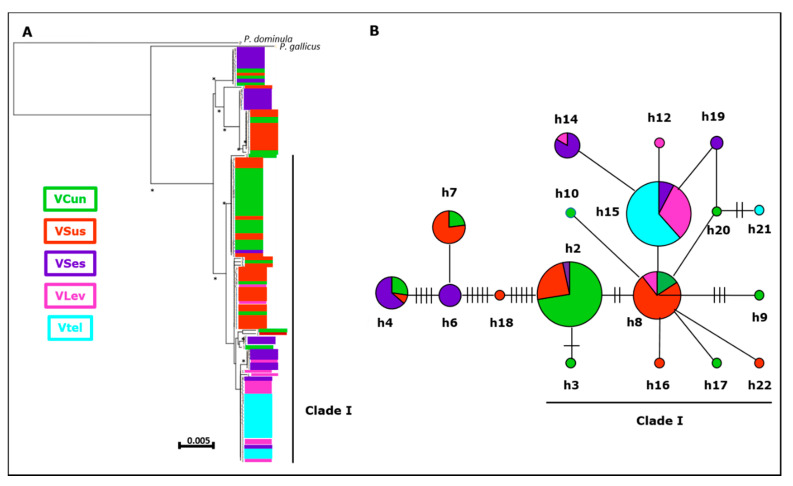
A schematic illustration of a neighbor-joining tree of *P. biglumis* mitochondrial sequences (**A**) and the minimum spanning haplotype network (**B**) based on them. The haplotypes are color coded at the region level as in Figure 1 and the largest clade I indicated with vertical (**A**) and horizontal (**B**) bars. Abbreviations of the regions are as in Appendix A. In (**A**), bootstrap values > 50% are indicated with an asterisk (*). In (**B**), the size of the symbols reflects the overall frequency of the haplotype and the ticks between the haplotypes are the number of nucleotide changes between the haplotypes (no ticks = one change, one tick = two changes, etc.). Note that the haplotype codes in *P. biglumis* (Figure 2) and *P. atrimandibularis* (Figure 3) do not refer to the same sequences.

**Figure 3 insects-11-00585-f003:**
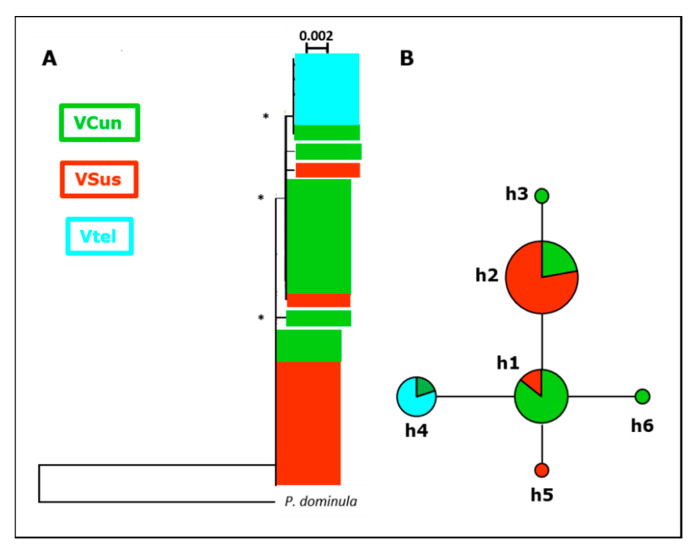
A schematic illustration of a neighbor-joining tree of *P. atrimandibularis* mitochondrial sequences (**A**) and the minimum spanning haplotype network (**B**) based on them. The haplotypes are color coded at the region level as in Figure 1. Abbreviations of the regions are as in Appendix A. In (**A**), bootstrap values > 50% are indicated with an asterisk (*). In (**B**), the size of the symbols reflects the overall frequency of the haplotype and the ticks between the haplotypes are the number of nucleotide changes between the haplotypes (no ticks = one change). Note that the haplotype codes in *P. biglumis* (Figure 2) and *P. atrimandibularis* (Figure 3) do not refer to the same sequences.

**Figure 4 insects-11-00585-f004:**
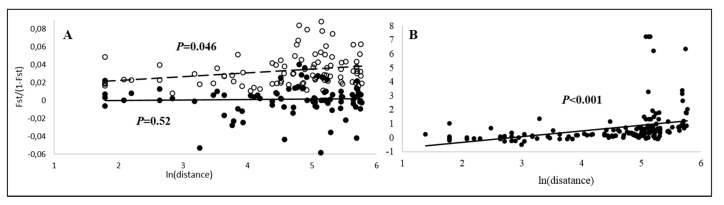
Isolation-by-distance in nuclear (**A**) and mitochondrial (**B**) markers in *P. biglumis*. In panel A, open circles and dashed line are for DNA microsatellites and closed circles and solid line for AFLP markers. In nuclear markers, only populations with *n* > 2 were included. Matrix correlation was significant in DNA microsatellite and mitochondrial markers, but not in AFLP markers.

**Table 1 insects-11-00585-t001:** Genetic diversity in *Polistes biglumis* and *Polistes atrimandibularis* populations in DNA microsatellite, amplified fragment length polymorphism (AFLP), and mitochondrial markers; *n* is the sample size for nuclear markers. *H*_E_ is expected heterozygosity according to Hardy-Weinberg equilibrium and *h*_AFLP_ haplotype diversity, both averaged across loci for each population; *h*_mt_ is haplotype diversity and π is nucleotide diversity. Values of *H*_E_ and *h*_AFLP_ and the average of *h*_mt_ and π in Valsesia were calculated for populations with sample size *n* > 2. *P. biglumis* populations parasitized by *P. atrimandibularis* are marked with an asterisk (*). Note that the haplotype codes in *P. biglumis* and *P. atrimandibularis* do not refer to the same haplotypes. Singleton haplotypes are in bold.

Population		DNA Microsatellites	AFLP	mtDNA		
	*n*	*H*_E_ [SE]	*h*_AFLP_ [SE]	Haplotypes Found	*h* _mt_	π [SD]
***P. biglumis***						
*Valli del Cuneese*						
Ferrere *		0.65 [0.07]	0.23 [0.02]	10*h2, h4, h7, h8, **h9**, **h10**	0.57	0.003 [0.002]
Terme di Valdieri	9	0.64 [0.09]	0.30 [0.03]	4*h2, **h3**, h4, h7, **h20**	0.80	0.004 [0.003]
Lago della Rovina	4	0.61 [0.08]	0.11 [0.03]	4*h2, h8	0.50	0.001 [0.001]
Fondovet	7	0.59 [0.11]	0.20 [0.02]	4*h2, h4, h7	0.60	0.004 [0.003]
Colle di Sampeyre	1	-	-	**h17**	-	-
Valle Pesio	1	-	-	h8	-	-
*Average*		0.62	0.21		0.61	0.003
*Val Susa*						
Montgenèvre *	48	0.69 [0.06]	0.17 [0.01]	3*h2, h4, 4*h7, 6*h8, **h16**, **h22**	0.80	0.004 [0.002]
Alpe Plane	16	0.68 [0.08]	0.12 [0.02]	2*h2, 2*h7, 5*h8, **h18**	0.73	0.004 [0.003]
Val di Thuras *	8	0.58 [0.12]	0.16 [0.02]	h2, 2*h7, h8	0.83	0.006 [0.004]
Cesana	9	0.62 [0.09]	0.15 [0.02]	h2, 2*h7, 2*h8	0.80	0.005 [0.003]
*Average*		0.64	0.15		0.79	0.005
*Valsesia*						
Frazione Dorf	5	0.70 [0.03]	0.15 [0.03]	h6, 2*h19	0.67	0.006 [0.005]
Rimella	11	0.64 [0.05]	0.25 [0.02]	h4, 4*h14, 2*h15	0.67	0.002 [0.002]
Fobello	2	-	-	h4, h6	1	0.005 [0.005]
Rima	2	-	-	h2, h6	1	0.007 [0.007]
Carcoforo	2	-	-	2*h4	0	0 [0]
Alpe di Mera	2	-	-	h4, h6	1	0.005 [0.005]
Sant’Antonio	9	0.70 [0.06]	0.20 [0.02]	2*h4, 2*h6, h14, h19	0.87	0.005 [0.003]
*Average*		0.68	0.20		0.80	0.005
*Val Leventina*	21					
Cari		0.69 [0.02]	0.19 [0.02]	2*h8, **h12**, h14, 8*h15	0.79	0.001 [0.001]
*Valtellina*						
Trivigno *	11	0.64 [0.07]	0.13 [0.02]	6*h16, **h21**	0.29	0.001 [0.001]
Campovecchio *	12	0.68 [0.06]	0.23 [0.02]	10*h16	0	0 [0]
*Average*		0.66	0.18		0.14	0.001
Average across all populations		0.65	0.19		0.66	0.002
***P. atrimandibularis***						
Valli del Cuneese	11	0.68 [0.04]	0.21 [0.02]	6*h1, 2*h2, **h3**, h4, **h6**	0.71	0.001 [0.001]
Val Susa	18	0.72 [0.06]	0.22 [0.02]	h1, 7*h2, **h5**	0.42	0.001 [0.001]
Valtellina	4	0.70 [0.04]	0.19 [0.03]	4*h4	0	0 [0]
Average across all populations		0.70	0.21		0.56	0.001

**Table 2 insects-11-00585-t002:** AMOVA results, with the proportion of genetic variation at different hierarchical levels (%) and the associated fixation indices for DNA microsatellites (*F*), AFLP, and mitochondrial markers (*Φ*). The ratio of mitochondrial to DNA microsatellite differentiation (*Φ*/*F*) is given when both values are positive. Note that AFLP markers and mitochondrial sequences are haploid and the within individual level is therefore substituted by the within-population/regions level. DNA microsatellite and AFLP analysis is based on 14 populations with *n* > 2, and mitochondrial analysis on 18 populations with *n* ≥ 2.

Source of Variation	DNA Microsatellites	AFLP	mtDNA	
%	*F*	*p*	%	*Φ*	*p*	%	*Φ*	*p*	*Φ* _mt_ */F*
***P. biglumis***/populations divided to regions
Among regions, *F*_CT_, *Φ_CT_*	2	0.017	0.001	12	0.118	0.001	28	0.280	0.001	16.5
Among pops within regions, *F*_SC_, *Φ*_SC_	0	−0.002	0.611	1	0.016	0.164	0	−0.020	0.693	
Among individuals within pops, *F*_IS_	10	0.104	0.001							
Within individuals, *F*_IT_	88	0.118	0.001							
Within populations				87			72			
***P. biglumis***/three regions shared by *P. biglumis* and *P. atrimandibularis*, no population level
Among regions, *F*_CT_, *Φ_CT_*	1	0.005	0.015	7	0.065	0.002	28	0.280	0.001	46.7
Among individuals within pops, *F*_IS_	9	0.093	0.001							
Within individuals, *F*_IT_	90	0.098	0.001							
Within populations				93			72			
***P. atrimandibularis***/three regions, no population level
Among regions, *F*_CT_, *Φ_CT_*	7	0.073	0.001	9	0.085	0.063	47	0.469	0.001	6.42
Among individuals within pops, *F*_IS_	10	0.113	0.002							
Within individuals, *F*_IT_	82	0,178	0.001							
Within regions				91			53

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
