# Peer review of "Strong Gene Flow Undermines Local Adaptations in a Host Parasite System"

_insects, 2020, doi:10.3390/insects11090585_

Round 1

Reviewer 1 Report

I enjoyed reading this manuscript. The work examines the population genetics of a host and parasite wasp in the Italian alps, with the authors discussing gene flow and the potential for adaptation and. Arms races between host and parasite. The molecular genetics approaches appear competently analysed. I think that there is sufficient data for a publication here, but would strongly suggest revising the framing of the work.

Specifically, the authors provide a data that allows conclusions to be made on the geographic structuring of the host species (in particular) and parasite (to some extent). I don’t believe, however, that much can be concluded about the adaptive potential of the hosts and parasites. The key issue is that the authors examine a small set of neutral markers. These markers allow us to consider geographic structuring but in all probability tell us nothing about local adaptation and adaptive potential. I think the title, abstract, introduction and discussion need to be heavily modified to acknowledge this issue.

Other specific comments:

Abstract: Please italicise the genus and species names, in addition to revising the work as suggested above.

Introduction, lines 110-116: I suggest making this ‘second goal’ of your study as the primary goal. I also think that you need to specifically acknowledge in the introduction the previous publication in this study system, using some of the samples, of Bonelli et al. (2015) ‘Population diversity in cuticular hydrocarbons and mtDNA in a mountain social wasp’ in the Journal of Chemical Ecology 41:22-31. In that paper it is concluded that “These results suggest that the populations of P. biglumis in the Alps are geographically isolated from one another, favoring their genetic and chemical differentiation”. That publication should be acknowledged and discussed in the discussion as well. How do the two papers differer in their conclusions?

Materials and Methods, lines 136-139: An unfortunate limitation in the study is the number of wasps used in the analysis of the parasite species. Only 33 individuals were used. It is thus hard to make a direct comparison with the host species with a sample size of 215. I think rarefaction such as in a haplotype discovery curve would be useful here to compare the two species.  

Materials and Methods, lines 168-171: There is a different font size used in this section. In marking student essays such a formatting change always flashes a warning sign for copying and pasting.

Results, Table 1: This table could go into the supplementary material. I think Figure 3 could go to the supplementary material as well. The map in figure 1 is OK, though I much preferred the map used in Bonelli et al. (2015). Journal of Chemical Ecology 41:22-31, which shows the specific location of the site in Europe. I’d urge the authors to place the figures and tables in a better order within the text.

Discussion, lines 390-396: Are there any studies of other animals or plants on genetic structuring within these study sites that could back up your conclusion? With regard to wasps, I think the authors should specifically discuss reference [80] to a greater extent.

Discussion, lines 486- 545: I think this section on “Assumptions revisited” should largely be removed or heavily down-sized. Focus on the geographic structuring.

Reviewer 2 Report

The authors of this manuscript sampled one alpine host species of a social parasitic wasp to conduct a population genetic study and investigate gene flow and local adaptation in both antagonists. As such data is rather rare and the research question a very interesting one I think this manuscript merits publication after some minor changes. There are still quite a few typoes and some sentences are very long. Please consider breaking them up. As this manuscript uses to some extent data from an already published paper, the authors should clearly delineate how the new study will add to the already existing one. In some part this is already done, but I would try to emphasise this a bit more in the introduction. Also in the discussion, the authors are overly negative about getting their predictions wrong and having underestimated gene flow. I suggest to re-phrase this in a more positive way. Also, the findings suggest that the authors should continue investigating this host-parasite interaction and probably extend it to all the host species of this parasitic wasp species. As I only have a few minor comments, I have directly added them to the PDF of the manuscript and uploaded it. I hope this is fine with the authors.

Round 2

Reviewer 1 Report

I've worked through the manuscript and changes made by the authors. There is a new analysis and the authors have modified the structure of the paper. I think it improved and is suitable for publication now. The paper, at least as it is presented to me, needs some small formatting changes but appears otherwise acceptable.